# Seismic monitoring of urban activity in Barcelona during COVID-19 lockdown

Jordi Diaz<sup>1</sup>, Mario Ruiz<sup>1</sup>, José-Antonio Jara<sup>2</sup>

<sup>1</sup> Geo3Bcn-CSIC, c/ Solé Sabarís sn, Barcelona, Spain
 <sup>2</sup> Institut Cartogràfic i Geològic de Catalunya, Barcelona, Spain

Correspondence to: Jordi Diaz (jdiaz@geo3bcn.csic.es)

Abstract. The city of Barcelona has been covered during the COVID-19 pandemic lockdown by a dense seismic network consisting of up to 19 seismic sensors. This network has provided an excellent tool to investigate in detail the background seismic noise variations associated to the lockdown measures. Permanent stations facilitate to compare the seismic noise

- recorded during the lockdown quieting with long-term variations due to holiday periods. On the other hand, the data acquired by the dense network show the differences between sites located near industrial areas, transportation hubs or residential areas. The results confirm that the quieting of human activity during lockdown has resulted in a reduction of seismic vibrations in the 2-20 Hz band clearly higher than during holiday seasons. This effect is observed throughout the city, but only those stations not affected by very proximal sources of vibration (construction sites, industries) are clearly correlated with the level of activity
- denoted by other indicators. Our contribution demonstrates that seismic amplitude variations can be used as a proxy for human activity in urban environments, providing details similar to those offered by other mobility indicators.

### **1** Introduction

Connecting students with seismology is a challenge in countries not affected by strong earthquakes. In order to mitigate this problem, the SANIMS research project included the deployment of a seismic network within the city of Barcelona, with most

- of the instruments installed in the facilities of secondary schools (Diaz et al., 2020). The network included up to 19 seismic sensors, distributed with an inter-station spacing of 2-3 km. The objective of the project was to acquire data of scientific interest to investigate the feasibility of ambient noise studies in urban environments, but also to promote the knowledge about seismology and Earth Sciences among high school students. The network was deployed in September 2019 and 50% of the sites were instrumented using the low-cost Raspberry Shake devices, which provide online access to the data in real time, thus
- facilitating the implication of the students. An unexpected result of this deployment has been the possibility to study in detail the seismic noise variations within the city of Barcelona during the quieting associated to the COVID-19 lockdown.

As background seismic vibrations (often referred to as "seismic noise") at frequencies above 1 Hz are mainly dominated by human activities (i.e. Díaz et al. 2017), lockdown measurements applied to mitigate the COVID-19 pandemic, including the

- suppression of industrial and commercial activities and movement restrictions, have had a clear effect on the seismic noise levels. This has been reported on a global scale by (Lecocq et al., 2020), showing how the progressive implementation of restrictions in different parts of the world could be monitored using publicly accessible seismic data. Several regional studies have documented these noise variations in China (Xiao et al., 2020), India (Somala, 2020), Brazil (Dias et al., 2020), northern Italy (Poli et al., 2020) or Sicily (Cannata et al., 2020). On a local scale, (Lindsey et al., 2020) have used a ~5 km long fiber-
- optic cable in Palo Alto (California, USA) to acquire Distributed Acoustic Sensing (DAS) data and analyze the seismic noise variations during COVID-19 pandemic. However, as far as we know, there are no studies available that explore variations in seismic noise within a large city with a space between sites on the order of 2-3 km. This contribution focus on analyzing such variations, which can be important depending on the specific location of each site. In addition, the availability of long-term data in the permanent stations makes possible to put in context the noise reduction due to the COVID-19 lockdown in
- Barcelona.

Following the increasing number of COVID-19 cases during February and early March, measures to mitigate the pandemic effects started in Barcelona Thursday the 13<sup>th</sup> March 2020, with the cancellation of all the face-to-face classes in schools and universities. Next day, the Prime minister declared the State of Alarm, and announced the first lockdown measures (Phase 1),

- effective from 00:00 of Sunday 15<sup>th</sup> March. Citizens were ordered to stay indoors, except for basic activities such as commuting to workplaces, buying food and medicine, or dealing with emergency situations. Stores, retail stores, cafes and restaurants were closed and remote work was recommended in all possible cases. Two weeks later, as of Sunday, March 29, the lockdown restrictions were reinforced with a generalized closure of the services, industry and construction activity. These strict measures, hereinafter referred to as "Phase 2", were applied for two weeks and, in practice, resulted in an almost total quarantine. On
- April 12, during the Easter holidays, these restrictions were lifted, reverting to the initial lockdown terms for three more weeks (Phase 1b). On May 4, the plan to ease the lockdown began. The so-called "de-escalation" included four phases, each of which had a minimum duration of two weeks. Citizens were allowed to go out for short walks, then shops and restaurants were reopened under restrictive measures and commercial and industrial activities were gradually resumed. However, face-to-face courses in schools and universities were suspended for the remainder of the 2019-20 academic year. Finally, the State of Alarm
- expired at midnight on Sunday, June 21, and the country began the so-called "new normal", which includes measures such as the obligation to wear a mask in public areas or restrictions on the occupation of shops and restaurants.

# 2 Data and Processing

Barcelona city center is instrumented by a permanent network made up of 3 accelerometers managed by the Institut Cartogràfic i Geològic de Catalunya and one broad-band station managed by the Geo3Bnc-CSIC institute, all of them integrated in the CA

- network (Institut Cartogràfic i Geològic de Catalunya, 2000). An additional accelerometer, is operational at the Fabra Observatory, located in the hills surrounding the city. The broad-band instrument is a Nanometrics Trillium T120P sensor, with flat response extended till periods of 120 s. The accelerometric stations are equipped with large dynamic range sensors and dataloggers.
- The SANIMS temporary array, active from September 2019 to September 2020, consisted of six three-component short period sensors (GeoSpace 2 Hz) with dedicated dataloggers and eight one-component Raspberry Shake seismometers equipped with 4.5 Hz sensors (<u>https://raspberryshake.org/products/raspberry-shake-1d/</u>). All the data was transmitted in near-real time using wired ethernet connections and processed in the Geo3Bcn data center. The investigated zone covers an area of approximately 10 km<sup>2</sup> within the city of Barcelona, although a couple of sites are located outside of the municipality. The geometry of the
- network has been chosen to sample the main geological units of the Barcelona area, with sites installed in the most recent terranes near the sea, above the quaternary materials of the Barcelona plain and in the hills where Paleozoic materials outcrops (Figure 1).

As a first processing stage, the instrumental response is removed following standard procedures and the data is referred to 75 seismic acceleration, expressed in nm/s<sup>2</sup>. Next, the frequency content of the seismic data is analyzed using the Power Spectra Density (PSD), which provides a good quantification of the energy levels recorded at the different frequency bands. PSD is calculated using a Obspy implementation (Krischer et al., 2015) of the classical PQLX ("IRIS- PASSCAL Quick Look eXtended") software (Mcnamara et al., 2009). The data is divided into 30-minute windows with 50% of overlap and the PSD of each window is computed using the Welch method. Following (Lecocq et al., 2020), the default PQLX parameters have

80 been modified slightly to improve the frequency resolution and increase the spectra dynamic. The corresponding spectrograms show the power of the seismic acceleration, expressed in decibels (dB) referred to  $1 \text{ m}^2/\text{s}^4/\text{Hz}$ . To make easier the comparison

with the spectrograms, the seismic noise variations are discussed in terms of power, without converting the data to acceleration, velocity or displacement.

## **3** Results

## 85 3.1 Identification of the frequency band of interest

To analyze the effects of the COVID-19 quieting on seismic data, we must first identify the frequency range where this effect is best identified by inspecting the spectrograms at each station. Figure 2 shows the spectrograms of four representative sites for the time period beginning three weeks before closure and ending three weeks after the end of the State of Alarm (February 24 - July 12, 2020). Supplementary figure S1 shows the spectrograms for the 19 stations available.

Spectrograms show that human related activity dominates above 2.0 Hz for most stations, confirming the previous results presented by Diaz et al (2017). Below 1 Hz, in the frequency range commonly known as the microseismic peak, the spectrograms show a large similarity between all the stations, as the origin of the signal is related to the interaction of oceanic

waves (i.e., (Díaz, 2016a). The energy reduction during the lockdown period can be identified for frequencies up to 20-25 Hz, although some sites also show a significant reduction in the 35-45 Hz band. Therefore, we will focus our research on the seismic signals within the 2-20 Hz band, calculating the averaged noise power within this band.

#### 3.2 Diurnal and weekly noise power variation patterns during normal activity.

- Before we get into the discussion on the lockdown effects, we will review the general trends in noise variation during periods of normal activity. Figure 3 shows the time variation of the power spectra between 2 and 20 Hz for the two-week interval just before the lockdown. The selected sites correspond to the ICJA broad-band station, located near one of the main road access to the city, an accelerometer installed in the historical city center (BAJU) and two short-period stations installed in secondary schools (R888C and R4B31), one of them in an area with industrial activity.
- The first observation is the clear difference between the noise power during the day and at night. This variation, observed systematically, has an average value close to 15 dB, although some of the station in high schools show variations greater than 20 dB, which denotes the great influence of students' activities. High energy levels are mostly observed between 7:00 and 19:00 official time, although in some places, particularly those located near the main entrances of the city, the period of high energy period begins earlier. The time interval with minimum energy is shorter, generally lasting around 5 hours, between
- midnight and 5:00 AM. Typically, the maximum power during Saturdays and Sundays are around 5 dB lower than during working days. Most sites show an additional reduction on Sundays around 2-3 dB. Minimum levels are generally observed during Sunday through Monday nights, while power during Friday and Saturday nights is often higher than during weekday nights, reflecting more nighttime activity in the city during the weekends.
- The graphs for each station display a wealth of details that reveal the activities near each site. Most sites show a fairly steep increase in noise in the morning, while the decline in the afternoon is more gradual, showing that many of the city's activities begin around 8 AM, while its hours of completion are less uniform (schools, offices and stores usually close between 4 PM and 9 PM). For some of the stations in high schools (i.e. Figure 3b) the maximum amplitudes are reached only during the morning, following the students' schedule. A curious example of citizen activities monitored by the variations in seismic noise
- can be identified at station ICJA (Figure 3a); the two peaks observed during the afternoon of 7/3/2020 correspond to people

arriving and leaving the Football Club Barcelona stadium for a Spanish league football match, played at 6 pm between FCB and Real Sociedad.

| network | station | pre/  | pre/  | pre/new   | site description                                    |
|---------|---------|-------|-------|-----------|-----------------------------------------------------|
| code    | code    | lock1 | lock2 | norm (dB) |                                                     |
|         |         | (dB)  | (dB)  |           |                                                     |
| CA      | ICJA    | 5     | 6     | 2.        | Geo3Bcn, University area.                           |
| CA      | MTJR    | 4     | 5     | 1         | ICGC site, Montjuic park                            |
| CA      | BAJU    | 4     | 5     | 2         | Administration building, center town                |
| CA      | BAIN    | 3     | 4     | 1         | Administration building, center town                |
| CA      | FBR     | 3     | 3.    | 0         | Fabra Observatory, Collserola hills                 |
| YS      | BALM    | 5     | 6     | 3         | Inst. Balmes high school, center town               |
| YS      | JPLA    | 12    | 13    | 10        | Inst. J. Pla high school, (Horta)                   |
| YS      | ISAB    | 2     | 4     | 1         | Inst. Infanta Isabel high school, (Sant Martí)      |
| YS      | CASN    | -     | -     | -         | Particular home, residential area                   |
| YS      | MILA    | 3     | 4     | 2         | Inst. Mila i Fontanals high school, (Raval)         |
| YS      | PORT    | 2     | 4     | 0         | Barcelona Port                                      |
| AM      | RDB03   | 8     | 9     | 5         | Sagrada Familia school (Sant Andreu)                |
| AM      | RD3AF   | 3.    | 4     | 2         | Inst. Montserrat high school (Sant Gervasi)         |
| AM      | RBE49   | 5     | 6     | 2         | Tecla Sala school (Hospitalet Llobregat)            |
| AM      | R888C   | 6.    | 6     | 3         | Inst. XXV Olimpiada high school, near Montjuic park |
| AM      | R59E2   | 4     | 4     | 2         | Maristes Les Corts high school (Sants)              |
| AM      | R4D07   | 5     | 6     | 2         | Inst. La Sedeta high school, (Eixample)             |
| AM      | R4B31   | 3     | 6     | 0         | Voramar school (Poble Nou)                          |
| AM      | R0A45   | 17    | 18    | 5         | Inst. Costa i Llobera high school (Collserola)      |

**Table 1:** Location of the seismic network stations and differences in power amplitude observed between the different lockdown phases.

# 3.3 Generic trends of power spectral variations during the COVID-19 lockdown.

- As discussed in section 3.1, the effect of the quieting during COVID-19 is best detected in the 2 20 Hz band. This effect is most marked during the more common working hours range, which can be assumed to be 9:00-19:00 in Barcelona. Therefore, we have used this frequency band and time range to calculate the daily mean values of the power spectral density. Figure 4 shows the results for all the available stations. The first three weeks, corresponding to the pre-lockdown period, are used as a reference and the green background shading marks the working days. As expected, weekends are clearly marked by an energy
- minimum. The most prominent observations are: i) The effects of the different lockdown stages are clearly identified in the data. ii) There are large variations in amplitude between sites. iii) the details of the amplitude changes differ significantly between groups of stations.

Most of the stations have maximum values around -90 / -95 dB. However, a couple of stations (PORT, R4B31) are much louder, with reference values between -80 and '90 dB. On the other hand, some of the sites within the city (JPLA, RD3AF, CASN) have low values around 105 / 110 dB. The minimum value is cheared for EBR, a station located in the Calleared

- CASN) have low values around -105 / -110 dB. The minimum value is observed for FBR, a station located in the Collserola hills that surround Barcelona, outside the city and relatively far from roads and train tracks, and R0A45, installed in a high school within the city but near to the Collserola hills.
- To obtain a global picture of the noise power variations in the city, we have calculated an average power amplitude profile for 145 the entire network. The profiles obtained for each station (Figure 4) have been normalized to the 0-1 range and the results averaged to obtain a mean profile (Figure 5) that can be considered as a summary of the results discussed in this contribution. Despite the large differences in values between the sites, these normalized profiles show that the temporal evolution of the background vibration in the 2-20 Hz band is consistent for most of the sites. During phase 1, the mean power level gradually decreases, to reach a minimum during the two-weeks period of maximum restrictions (phase 2) and then increases smoothly
- until mid-July, without reaching pre-lockdown values. The reduction is also clearly shown during the weekends (minimum values in Figure 5) and in the difference between weekdays and weekends, which is minimal during the second week of phase 2.
- To take advantage of the large number of stations available, we have represented the same information as daily maps. The normalized values are located on a city map, gridded using a near neighbor algorithm and presented as supplementary figure 2. Figure 6 summarizes the information showing the images of four Mondays, each corresponding to one of the lockdown phases. Although the maps may include some spurious effects due to the interpolation procedure, they provide visual information on the variation of the seismic power, denoting clearly differenced pre-lockdown, lockdown phases 1 and 2 and lifting periods. In pre-lockdown times, most of the stations have their highest absolute values, thus appearing in the maps with
- normalized values close to 1 (Figure 6a). During lockdown phases 1 and 1b most of the sites remain below average, except for a couple of them that are affected by local sources of noise that will be discussed in the next section (Figure 6b). During the weeks with stricter lockdown measures (Figure 6c) all stations have the lowest power and the normalized values are close to zero. During the lifting period (Figure 6 d) the sites progressively increase in amplitude. Note that weekends and holidays (April 13, May 1, June 1, June 24) are identified in the daily snapshots by their low power levels (Suppl. figure S2).

## 3.4 Site-dependent characteristics of power variations during the COVID-19 lockdown

The relatively large number of sites within Barcelona makes it possible to analyze in detail the variations of power and to study the influence of very local noise sources, which can eventually distort the interpretation of the results. Figure 7 shows the power variations in the 2-20 Hz band for some selected sites. Suppl. Fig. S3 presents the same information for all the sites. In

- these figures, the light blue lines show the power amplitude evaluated every 15 minutes and the dark blue ones show the daily average value during business hours (9:00-19:00, official time). The orange lines show the mean value of the business hours amplitudes for an interval of one week, smoothing out the week-end lows and facilitating the discuss of the long-term variations.
- The drop in noise power following the onset of the quieting measures has a mean value of 4 dB during the phase 1 of lockdown (Table 1). The reduction is more prominent for those sites located in schools, reaching in some cases values up to 17 dB. This large reduction reveals the impact of very close sources of vibration (in this case, the movement of students in school) on the seismic data. During the stricter phase 2 lockdown period, one additional dB of reduction is observed for most of the sites, although in a couple of cases the decrease is greater. The reduction also affects periods with low levels, such as the weekends

(lows in the dark blue lines) or night periods (lows in the light blue lines). During lockdown, the difference in energy levels between weekdays and weekends is also drastically reduced. While in the days before lockdown this difference is around 5 dB, at the peak of the quieting it remains between 2 and 3 dB. As discussed above, most of the stations share a fairly similar pattern of variation, but some of them have clearly different results. Figure 7 displays the variation of power for several sites showing the general variation trend (Figures 7a-d) and for most of the sites with particular characteristics. (Figures 7e-f).

The ICJA station (Figure 7a) is installed in the basement of the Geo3Bcn institute, located within a university campus and near one of the main road entrances to the city. Diaz et al. 2017 have documented a close relationship between seismic noise around 10 Hz and the traffic entering and exiting from the city. The changes in seismic power are smooth, with a progressive decrease during the first two weeks and a clear minimum during the phase 2 lockdown. During phase 1b, the increase is constant and

- continues during the period of lifting. Station RBE49 (Figure 7b) is located in the Tecla Sala school, in the center of Hospitalet de Llobregat, an independent city forming a continuous urban environment with Barcelona. As with all stations located in schools, student activity is a major source of vibrations around 10 Hz. This is evidenced here by the lower noise observed the Friday, March 13, just before the start of the lockdown, when schools were already closed. The pattern is similar to that of ICJA, although a particular increase can be observed during the first week after the start of the release measures. Face-to-face
- classes did not restart in Barcelona until September 2020, but the increase probably reflects the activity of the school to prepare online courses and materials. The BALM seismometer (Figure 7c) is installed also in a high school, located in this case in the center of Barcelona, in the "Eixample" neighborhood and close to monuments such as Casa Batlló and Casa Milà. The seismic results show here a greater difference between the lockdown phases 1 and 2, suggesting that the effect of strict closure was more effective in the city center. The last example of sites that register quite similar noise variations is BAJU (Figure 7d), an
- accelerometer located in an administrative building in the historical center of the city, near shopping centers and tourist attractions. The results are consistent with previously commented examples, although some specificities can be highlighted. For example, the noise level is not stable during the two weeks of the phase 2 period, with the second one clearly quieter. Also note that the difference between weekdays and weekends and between day and night appears to be narrower than for the previous cases.

If we focus on stations having abnormal results, we can first comment on the results of station R4B31, located in the Voramar school in the Poble Nou area, on an area of old wetlands and swamps close to the beach (Figure 7e). The noise reduction during the lockdown phase 1 is modest, despite the fact that students stay at home. The only time period with a large noise reduction is during lockdown phase 2 and, in particular, during the second week of this phase. Also note that during phase 1b

- and during the lifting measures period the power reaches values that are very similar to those of the pre-lockdown period, contrary to what is observed in most sites. To understand these results, we visited the area and noticed that a historical industrial factory, "Industries Waldes" is located right in front of the school. This factory, still in full activity, is specialized in the production of buttons, snaps, metal fittings and other metal accessories and uses heavy machinery (<u>https://waldes.es/en/company/</u>). Following the government's guidelines, the industry only closed during the phase 2 of the
- lockdown, explaining the reduction in seismic noise during this period. Noise levels are also abnormally high in mid-June and mid-July, during a period with almost normal levels of urban activity. This seems to be related to the activity in a large construction site located about 150 m of the sensor. Station PORT, located next to one on the main container terminal in the port of Barcelona and at approximately 1 km of one of the largest water treatment plants in Europe (https://www.amb.cat/en/web/ecologia/aigua/instalacions-i-equipaments/detall/-/equipament/edar-del-prat-de-
- <u>llobregat/276285/11818</u>), seems also to be dominated by near sources of noise related to industrial activities, resulting is a noise reduction limited to the two weeks of total closure of activity (Suppl. Fig. S3). The results for these sites show that vibration sources located near the seismic sensor can dominate the signal and disturb the interpretation of the data in terms of

general human activity. Station CASN (Figure 7f) was installed in the basement of a private house located in the Paleozoic hills near Park Güell, relatively far from large avenues and industrial areas. In this case, the seismic noise does not reproduce

- clearly the lockdown quieting, although the weekly mean value (orange line) shows a small decrease during phases 1 and 2 and a slight but constant increase thereafter. The sharp change observed in most sites is not observed here. We think that the increase of the vibrations generated by the family members, usually working or studying at other places, but staying home since the beginning of the lockdown, compensates the general decrease in seismic noise, thus masking the change related to the start of lockdown. It can be noted that similar observations are been reported in social media by users of the Raspberry
- Shake citizen seismic network. Figure 7g shows the results of the Inst. Josep Pla high school, located in the Horta neighborhood. Here, contrary to the situation described for R4B31, the influence of students in seismic noise is enormous, as reflected by the very sharp decrease as of Friday, March 13. The noise level during business hours decreases by more than 12 dB and remains 10 dB below the usual values until the end of the investigated time period. This is interpreted as due to the quiet environment around the site, away from large avenues and industries axes and next to a medium-sized urban park. Finally,
- the data acquired at the FBR station (Figure 7h), located on the hills outside of the city, shows that, although a decrease in the weekly averages can be observed following the beginning of lockdown, the variations of power follow a clearly different pattern than for most of the sites. The spectrogram for this station (Suppl. Fig. 1) shows that the seismic noise variation in the 2-20 Hz band is very similar to that observed for frequencies below 1 Hz, a range for where wind and oceanic waves are widely admitted that the dominant sources of noise (i.e. (Díaz, 2016b). Therefore, we conclude that for this station, with low levels of
- human activity nearby, the lower frequency seismic sources extend to the 2-20 Hz band.

To better discuss the differences between the different stations, we have mapped the absolute power values in the 2-20 Hz range, following the same approach discussed previously for the normalized data. Supplementary Figure S4 show all the snapshots, while Figure 8 shows the images corresponding to four Mondays at different lockdown phases. During the pre-

- lockdown period (Figure 8a) there is a rough correlation between the seismic noise and the geologic zonation, with lower noise for stations located in Precambrian hills and higher levels in the sites located in the recent Quaternary sediments. However, this correlation has to be taken with caution, as the Quaternary areas include most of the city center, the area most affected by human activity. After the beginning of the restrictive measures, (Figure 8b), the noise distribution changes slightly, with the area with low noise levels extending towards the city center. The reddish spot near the coast correspond to station R4B31,
- affected by an industrial factory located nearby, and located over the most recent Holocene sediments. During the stricter lockdown interval (Figure 8c) the activity at this factory was suppressed and this is reflected in the seismic noise maps, which can be more directly related to geology. Figures8b and 8c show a lower amplification in Miocene areas around Montjuic than in the zone covered by quaternary sediments. The maps during lockdown phase 2 are consistent with the microzonation map proposed by (Cid et al., 2001) and with the results based on the analysis of fundamental resonance frequency using the
- horizontal to vertical spectral ratio (HVSR) by (Cadet et al., 2011). During the lifting period (Fig 8d) the vibrations affecting R4B31 reappears and noise levels progressively recovers, although without reaching the pre-lockdown levels. The area with lowest noise progressively reduces, although it remains larger than in pre-lockdown times. The snapshots in Supplementary Figure S4 show also the amplitude decrease during weekends and holidays. From this analysis of the geographical variation of PSD we can document, during the stricter lockdown period, the direct relationship between of the geological cover and the
- seismic amplification, without needing to use classical methods based on HVSR or tomographic inversions. However, this direct relationship becomes less clear during normal human activity periods, as industries and traffic close to each station strongly disturb the seismic power.

#### 3.6 Long term noise power variations

To put the decrease of seismic noise into perspective, we have calculated the power variation since January 2019 for three 265 permanent seismic stations located within Barcelona, the ICJA broad-band station and the BAIN and BAJU accelerometers (Fig. 9). As for Figure 6, light blue lines show data every 15 minutes, dark blue line represent the average during working hours and orange line show the weekly average during working hours. In this case, the data in the figure extends till the 30th September 2020. The spikes often observed for the BAIN and BAJU sites are related to data transmission problems and are not relevant for the discussion.

All three sites show a similar pattern, although small differences can be identified. During the Easter holidays of 2019, the average weekly noise decreases for all the sites, but is particularly visible at ICJA station, where the decrease reaches 4 dB. The 2019 summer holidays are clearly identified in the seismic data by a decrease throughout the month of August, the traditional period of holidays in Spain. This quieting is a little less marked than during Easter holidays but it last for a period

- of four weeks. During both holiday periods, BAJU shows a less prominent decrease than the other stations. This site is located in tourist area of the city center and we can hypothesize that the increase of tourist activity during holidays can offset the decrease in the activity of local citizens. The 2019 Christmas period is also well reflected in the seismic data, with a decline lasting about 2 weeks and reaching similar values that during the Easter holidays. A significant increase is observed at BAJU at the beginning of November 2019. Although we do not have specific information, this increase has not been observed for
- the rest of sites and is probably be related to civil works near to the station site.

As clearly shown in the data presented in Figure 9, the effects of the COVID-19 lockdown have a much greater impact on the seismic power of the 2-20 Hz band than the usual holiday seasons. During lockdown phase 1 the decrease is only slightly greater than during holidays, but during the two weeks with stricter closure, the maximum noise levels during business hours

are clearly below the levels observed during holidays and weekends. As mentioned early, during the release period, the power increases smoothly, but does not reach the pre-lockdown levels. During 2020, the Easter holidays where coincident with the lockdown period and could not be clearly identified in the data. On contrary, even in the context of limited citizen activity, the August holidays have a clear impact in the seismic noise levels.

## 4 Discussion and conclusions

- From the data analyzed in the previous sections, it is clear that seismic power in the 2-20 Hz band can be used as a proxy to monitor the level of anthropogenic activity in an urban environment. Verifying mobility patterns during lockdown periods is an important tool to analyze the efficacy of the different lockdown strategies adopted by civil authorities. The exceptionality of this period has led several large companies such as Google, Apple or Facebook to agree to make their mobility data public for a limited period. Similarly, the main mobile phone operators have shared data related to the mobility of the devices under
- their control with government agencies. Other mobile indicators can be the number of ticket validations in public transport systems and the estimations of vehicle movements in the city, obtained from vehicle counters installed in representative locations. In order to evaluate the real usefulness of seismic data to be used as a monitoring tool, we have compared our results with some of these indicators.
- Some of the indicators provided by large communication companies as Google or Apple are shown in Figure 10a together with the averaged seismic data. The indicators are normalized to the 0-1 range to make a visual comparison possible. The black line shows the data provided by Apple (<u>https://covid19.apple.com/mobility</u>) regarding the relative number of address requests on Apple Maps from people driving vehicles around the Barcelona area, compared to January 13, 2020 (Monday). Apple data shows a sharp decline during the first week of phase 1 and remains fairly stable until the beginning of phase 1b.

- During phase 1b and the lifting period, the indicator increases at a increasing rate. In mid-July, the indicator returned to prelockdown levels. The green line in Figure 10a shows the change in activity in places related to public transportation such as metro, bus and train stations, as provided by Google for the Barcelona area (<u>https://www.google.com/covid19/mobility/</u>). These data refer to the median value for the same day of the week during January 2020 and hence does not show the weekdays / weekends variation. During lockdown phases 1, 2 and 1b, the pattern is similar to the Apple's data. However, during the first
- 310 weeks of the lifting period, the increase is more marked. In contrast, during the "new normal" period, the indicator remains fairly constant.

Turning now to the datasets made available through public institutions, we have first analyzed the data provided by the Barcelona transport authority (https://www.atm.cat/web/ca/covid19.php) regarding the number of validations of trips in public

transportation systems within the city, including bus, metro and tramway (blue line in Figure 10b). This dataset is not based in mobile phone activity but comes directly from transportation operators. The absolute number of daily validations ranges from a low of 72.797 on April 12, during the second week of phase 2 lockdown, to 1.893.739 on July 9. As in previous cases, the data has been normalized to the 0-1 range. As in previous cases, the use of public transportation begins to increase during Phase 1b, but in this case the data shows a very constant rate, both on weekdays and on weekends, until mid-July.

Finally, we have inspected the data provided by the Spanish Institute of Statistics (INE, <u>https://www.ine.es/covid/tabla.htm</u>) on the mobility at the scale of the small zones, chosen to include around 20 000 inhabitants each (in the case of Barcelona, the this means more than 70 zones). The data are based on the information provided by the three main mobile phone operators in Spain on the location of mobile phone devices, and show the percent of population moving from their home during each day

- as is referred to the value between the 18<sup>th</sup> and 21<sup>th</sup> November 2019 (Monday to Thursday). Only the data corresponding to the lockdown period has been opened. We have calculated the mean values of this data for all the mobility areas of Barcelona and represented them as a green line in Figure 10b, once normalized to the 0-1 range. As expected, the overall correlation between the different indicators is good, all of them highlighting the decrease in activity during the lockdown period. However, some significant differences can be observed between them. The seismic data clearly show the decrease in activity during
- phase 1 and phase 2, with the minimum level observed during the second week of phase 2. During phase 1b, the average seismic amplitude quickly recovers the levels of phase 1, a fact that is also observed in the INE mobility data and in the number of public transport validations (Figure 10b) but not so clearly in the data from Apple and Google. The main differences between the different indicators appear during the end of the release phase and the beginning of the "new normal" period. Public transportation validations increase at a fairly constant rate from the end of March to mid-July, while Google's public
- transportation estimations show a marked increase at the beginning of phase 1b and stabilize from the end of June. When comparing the INE mobility estimations and Apple's estimate for driving activity, significant differences can be identified, for example, when comparing between weekdays and weekends. Apple's data for July has a maximum value on Friday that does not appear in other indicators, such as the volume of traffic in and around the city. Therefore, each indicator has its own peculiarities and none of them can be considered as a ground-truth reference.

The most relevant difference between the seismic data and other indicators is the fact that seismic power has remained almost stable since the end of May and has not recovered its pre-lockdown level, while the other indicators continue to increase during June and July until they recover their levels. This fact can be understood by assuming that seismic noise arises from two main sources, traffic and movement in the vicinity of the sensor, normally due to the activity of people in the building, such as

students in high school or workers in the office. This hypothesis is confirmed by comparing the seismic noise level at each site with the INE data for each mobility zone. Figure 10c shows two clearly different cases. Station R4D07, located in a secondary school near Sagrada Familia, shows a good match between mobility and seismic data during phases 1, 2 and 1b, but afterwards

mobility continues increasing while seismic remains stable. On the other hand, in the ICJA station the seismic and mobility indicators have a more similar evolution, although the difference between weekdays and weekends is greater for the seismic

- data during the lockdown period. This site is located near one of the main access roads to Barcelona, a fact that may explain the greater relative contribution of traffic. From these observations, we conclude that background seismic vibrations in the 2-20 Hz band are related by the traffic volume only up to a certain threshold, above which further increases in traffic do not affect the seismic data. The second main source of vibrations in this band is related to human activity near the seismometer. The daily variations of seismic amplitude in this frequency band does not correlate with the subway schedule during the
- pandemic, which confirms previous contributions where it was stated that the subway transportation system produces vibrations at frequencies ranging mostly between 20 and 40 Hz (Díaz et al., 2017; Green et al., 2017; Hinzen, 2014).

As a summary, we can conclude that seismic data in the 2-20 Hz band are a good proxy of the local anthropogenic activity. The effects of the lockdown measures taken after the outbreak of the COVID-19 pandemic are clearly seen at most sites, as

- previously documented at worldwide scale (i.e. (Lecocq et al., 2020). The variations in seismic power over time make it possible to distinguish the different phases of lockdown, at least in a similar way to other mobility indicators. Comparing with previous data acquired by the permanent seismic stations in Barcelona, we can conclude that the quieting during lockdown is clearly higher than the noise lowering observed during holiday periods. Having a dense seismic network deployed over the city has provided the opportunity to analyze in detail the effects of the lockdown measures on the 19 available sites, showing
- that the mean results reproduce accurately the human activity pattern, but also that significantly different results are obtained for some sites, where near and specific noise sources dominate the recorded signal. Seismic noise between 2 and 20 Hz is mostly sensitive to traffic, industrial and construction activities at distances of few hundred meters and can hence be considered as a proxy to monitor these human activities. Our results suggest that above a certain threshold, the increase in traffic is no longer reflected in the seismic data. This threshold value is probably related to the distance between the seismometer and the
- source area and to the attenuation properties of the subsoil. Compared to other mobility indicators, seismic data have some important advantages; the data is open and its availability does not depend of the decisions of private companies, traffic and industrial activities are aggregated, the installation and maintenance of the sensors is easy, and there are well-established protocols to share the data in nearly real-time. This opens the door to the systematic use of seismic sensors as tools to monitor human activity in urban areas. This monitoring can be crucial in times of crisis that involve changes in the usual mobility
- patterns, but it can also be of practical use in regular times.

We want also to highlight the value of operating an urban seismic network for the dissemination of Earth sciences, in particular if some of the stations are installed in secondary schools. Having these instruments available in the classroom is a great opportunity to develop students' interest not only in seismology or Earth Sciences, but also, more generally, to promote the

- use of scientific methodologies. Collaboration with educational centers has led to three high school research projects based on the analysis of seismic data, despite the scarce presence of Earth sciences in Spanish curricula. Seismic records in urban environments also provide an opportunity to access the media and have a significant impact on social networks; the COVID-19 lockdown has been an extraordinary opportunity in this regard, but even at regular times, the detection of sporting or musical events, extreme weather phenomena, controlled explosions, disturbances in the public transport system, etc., can be of interest
- to a public unfamiliar with Earth Sciences.

## Author contribution

JD, MR and JAJ worked on the data acquisition, processing and analysis. JD wrote the manuscript and prepared the figures, with contributions from all co-authors.

## **Competing interest**

The authors declare that they have no conflict of interest.

#### Acknowledgments

We want to thank the secondary school teachers involved in this research project for their collaboration and help: Xavier Valbuena (Escola Tecla Sala), Antoni Cirera (Inst. Costa Llobera), Nuria Prat (Inst. Balmes), Xavier Pie (Inst. Infanta Isabel d'Aragó), Miguel Manzano (Inst. Montserrat), Gemma Briones (Escola Voramar), Oscar Arribas (Inst. Milà i Fontanals),

- Robert Barriche (Escola Sagrada Familia, Sant Andreu), Elisa Rodriguez (Inst. Josep Pla), Angel Mota (Inst. La Sedeta) and Carles Sanchiz (Inst. XXV Olimpiada). We also want to thank Dani Ruiz, Jordi Vila and Joaquim Cortés, from "Dep. de Medi Ambient" of the "Port de Barcelona" for facilitating the installation of one of the sites. Finally, we thank the "National Institute of Statistics" (INE) and the "Autoritat del Transport Metropolità de l'àrea de Barcelona" (ATM) for sharing detailed mobility data.

420

This work has benefited from open source initiatives such as Obspy (Krischer et al. 2015) and GMT (Wessel, P., Smith, W.H.F., Scharroo, R., Luis, J., Wobb 2013). Data analysis has been done using the SeismoRMS code kindly distributed by Thomas Leccoq (https://github.com/ThomasLeccoq/SeismoRMS).

This is a contribution from the SANIMS project (RTI2018-095594-B-I00), funded by the Ministry of Science, Innovation and Universities of Spain, with the additional support of the 2017SGR1022 grant from the Generalitat de Catalunya.

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
