# Peer review of "Seismic monitoring of urban activity in Barcelona during COVID-19 lockdown"

_Solid Earth, 2020_

## Referee Comment (RC1) · Andrea Cannata (Referee) · 8 Dec 2020

The paper "Seismic monitoring of urban activity in Barcelona during COVID-19 lockdown" by Jordi Diaz and co-authors deals with the seismic signature of the lockdown measures as observed by a very dense local seismic network, installed in Barcelona and composed of a fairly wide variety of instruments: broad-band sensors, short-period sensors, accelerometers and Raspberry shake seismometers. The paper is in a very good shape and scientifically sound, it shows very interesting results regarding how the amount of seismic noise reduction, due to the lockdown measures, is strongly site-dependant even in a so dense network and reflects the local human activity variations. In my opinion, the manuscript deserves to be published in Solid-Earth after minor revision.

[Figure]

MAJOR POINTS

- Section 3.1: it is not clear how you identified the frequency band of interest, that is, the band where the seismic signature of lockdown measures is most evident. It is hard to identify it just by looking at the spectrograms. In addition, each spectrogram shows peculiar features, different from the others.

- Line 175: how did you calculate such mean value, as well as the power values shown in Table 1? By taking into account the whole time series, or by focusing on week-days/daytime?

- The "Discussion and conclusions" section does not discuss all the findings of the manuscript, but it mainly focuses on the comparison between seismic data and mobility information. I suggest to rename the section, and write another section, truly discussing all the findings of the paper.

MINOR POINTS

- Line 59: What does "CA" indicate?

- Lines 58-72: the sampling rate information is missing for all the instruments.

- Line 78-79: please provide further information about the spectral analysis. For instance, did you divide the 30-min-long windows into smaller windows? If so, how long do the smaller windows last?

- Line 80: What do you mean when you talk about "spectra dynamic". Sorry, I do not know this term.

- Line 95: "(i.e., (Díaz, 2016a)": please remove the double brackets.

- Line 108: is "official time" the local time?

- Lines 108-109: "the period of high energy period begins earlier": sounds a bit strange...

- Line 126: "Location of the seismic network": actually, the location is not indicated in the table, but rather site descriptions and power amplitude information.

- Line 184: "characteristics. (Figures 7e-f)" –> "characteristics (Figures 7e-h)"?

- Line 234-240: the power level at this station is very very low compared to what is recorded by the other sensors (it is hard to clearly read it, but it seems to be lower than -300 dB). Is it reasonable? Or is there a problem in the instrument?

- Line 258-262: this finding is very interesting. However, it is not highlighted in the abstract, neither in the Discussion and conclusions section.

FIGURES

- Figure 1: it would be useful to add in the legend information about the symbols used for the seismic stations. In addition, I suggest to use different symbols (or colors) for short-period and Raspberry Shake sensors (as far as I understand they are both indicated by red dots). The font size of the legend is very small, I suggest to make it bigger.

- Figure 2: it would be useful to indicate the different phase names in the figure (similar to what has been done in Figures 4 and 5).

- Figure 3: I suggest to increase the font size. In addition, please correct the caption "Data is expressed as dB as dB relative..." –> "Data is expressed as dB relative..."

- Figure 4: it is really hard to identify the names of the stations associated with each time series. I suggest to increase the font size of the legend, and to sort the names in the legend into descending noise power order.

- Figure 6: I suggest to increase the font size of the labels surrounding each plot.

- Figure 7: I suggest to increase the font size and to indicate the different phase names in the figure (similar to what has been done in Figures 4 and 5).

[Figure]

- Figure 8: I suggest to increase the font size of the labels surrounding each plot. In addition, to make the comparison between seismic noise power and geology easier, I suggest to add another subplot with a schematic map showing the main material distribution, such as Paleozoic, Holocene and Pleistocene materials (a sort of simplified version of Figure 1), as well as the location of the places you cite in the text, as the city center, the industrial factory affecting the station R4B31, Montjuic.

- Figure 9: "Eastern 2019" –> "Easter 2019".

- Figure 10: I suggest to increase the font size and to indicate the different phase names in the figure (similar to what has been done in Figures 4 and 5).

———————————————————————

---

## Referee Comment (RC2) · Koen Van Noten (Referee) · 25 Jan 2021

The authors present a very nice local study on the lockdown effects in Barcelona observed on the fixed seismometers and (school) Raspberry shakes. In general this is a clear, well-written study which reads fluently. In this review I suggest only minor changes and propose only cosmetic changes and corrections.

There is one statement though that the authors did not fully discuss. The authors explain that they see a direct relationship between the geological cover and the seismic amplification and that this can be detected without the need of performing a microzonation or tomographic inversions. It is somewhat difficult to compare the microzonation results with your findings. It would help to add the zoning of Cid et al. (2001) on the

maps in Fig. 8 to see correspondence. Also, why is this correspondence there? Because of interference in the subsoil in this frequency band? This needs some more explanation.

**Minor comments:**

L22: ambient noise studies: this is a bit vague? What purpose have these ambient noise studies?

L31: change brackets to Lecocq et al. (2020); same for L34: Lindsey et al. (2020) + Check the rest of the paper. Authors are often within the brackets when they are an active part of the sentence. I refer to this comment by writing "brackets" in this review.

L34-35: "However, as far as we know, there are no studies available that explore variations in seismic noise within a large city with a space between sites on the order of 2-3 km." Be careful here: you mean that really no-one studied ambient noise variations? There are a lot of geophysical papers about measuring noise in a city, but these are mostly used for array processing and subsurface identification, nevertheless they use the noise variations.

L43. Thursday the 13th of March

L45. Sunday 15 March or Sunday 15th of March
In this paper be consistent how you refer to dates. I already noticed 3 different

notations. So use one consistent notation: e.g.
Sunday 15 March
Sunday 15th of March
Sunday, March 15 (probably the last is the best choice, as you continue using this notation, e.g. March 29, May 4, etc. . .)

L67: Is there an official Raspberry shake paper to refer to instead of using the weblink? + It would not harm to cite few city-context papers where Raspberry Shake was already successfully used (e.g. Anthony et al 2018, SRL; De Plaen et al. 2020 (this special issue volume))

L69: outside the municipality

L71: "most recent terranes near the sea": give a geological time frame. Holocene? Pleistocene?

L71: Is "materials" a correct term? hard rock?

L89: "brackets"

L95: "brackets"

L96: a sentence is missing to link L95 and L96, something like: "because not all stations show reduction up to 45Hz, we will only focus our research..."

L106: although some of the stations

L117: "its hours of completion": weird phrasing. If you refer to the activities, it should be "their hours of completion"; not sure if completion is the correct word to use.

L119: " a curious observation": is it "curious" when you explain what it is? Perhaps use "remarkable" or "notable", . . . ?

Table 1: This table should be self-standing. Explain what the pre/lock1, pre/lock2 , pre/new norm columns stand for.

L135: "clearly identified": try to minimize the use of words as "clearly, obvious" etc, rather explain why it's clear. E.g. a drop follows an lockdown phase, a mean rise follows a phase, etc. . .

L139: '90dB = -90 dB ?

L150: clearly is ok in this phrase

L155: nearest neighbor algorithm

L170: here again reference should be made to the technique how these figures were computed. Or this can be mentioned in the caption of Fig. 7

L175: indicate which stations you are talking about by e.g. putting station names in brackets behind "in schools"

L186: (2017) "brackets"

L239: "brackets"

L252 : for people that never visited Barcelona, indicate where montjuic is located

L253: how are both maps consistent? Please explain. Do the 4 zones correspond to the microzonation map of Cid2001? Perhaps it would be interesting to put the zonation boundaries on the map.

L254: "brackets"

L255: "brackets"

L280: "probably be related to civil works near to the station site." This is speculation. Needed in this paper? You can just say this increase remains unexplained due to lack of local site/communal information.

L358: I think you can separate the discussion (everything before L358) and the conclusion (starting from L358)

L358: in the discussion, a discussion on the comparison between your findings and microzonation findings is lacking. Is this anywhere else observed? Can this be done in other cities (check e.g. papers in this special issue). If this is new, it should be

mentioned in the conclusions, and perhaps in the abstract.

L360: "brackets"

L373: I think it's "near real-time", not nearly real-time. Please check.

L401-403: add these cited references to the reference list and follow the proper citation rules in these lines: i.e. Wessel et al. (2013), Lecocq et al. 2020.

T. Lecocq, F. Massin, C. Satriano, M. Vanstone, T. Megies, SeismoRMS - A simple Python/Jupyter Notebook package for studying seismic noise changes, Version 1.0, Zenodo (2020); doi:10.5281/zenodo.3820046

L437: wrong author list: either provide full names of all 75 authors or write: Lecocq, T., Hicks, S. P., Van Noten, K., Van Wijk, K., Koelemeijer, P., et al. : Global quieting of high-frequency seismic noise due to COVID-19 pandemic lockdown measures, Science (80)., 369(11 September), 1338–1343, 2020.

**Figures:**
Concerning the figures, the maps are often of low resolution and fonts and legends are too small. Please check if this is due to the conversion to pdf, or if you indeed provided low resolution maps. If so, increase resolution.

**Figure 1**:
- The colors of the geology in the background of the Holocene, Pleistocene, Pliocene

and Carboniferous periods are very difficult to distinguish.

- Add coordinates to the figure.

- A little inset with the locality of Barcelona indicate on the scale of Spain would be of interest for the international reader.

- can you use a different symbol for Seismometers and Accelerometers (or Raspberryshake)?

- caption: "The dark and light blue dots show the permanent broad-band and accelerometric stations, respectively (?).

**Figure 2**: in the caption, please write again the lockdown phase dates that correspond to the 3 dashed lines.

**Figure 3:**

- indicate when lockdown happened with a vertical bar on this figure

- add to caption: Weekdays are indicated in green.

- add to caption: Trends in noise variation prior to lockdown.

- explain again where the stations are located (e.g. AM.R888C in school, . . .)

- What are the spikes on CA.BAJU ?

- Refer to Lecocq's seismo RMS technique either in the text (L102) or in the caption, to highlight how you made this figure:

T. Lecocq, F. Massin, C. Satriano, M. Vanstone, T. Megies, SeismoRMS - A simple Python/Jupyter Notebook package for studying seismic noise changes, Version 1.0, Zenodo (2020); .doi:10.5281/zenodo.3820046

**Figure 4**: It would be more intuitive if the legend could be ordered according to the observations: so MTJR on top (dark green), then R4B31 (light green), etc. . . this would easier read the diagram.

[Figure]

**Figure 6:**
- This figure is of low resolution. Can you increase the resolution.
- Also the coordinates are unreadable
- Topographical contours are included, but height is not shown in the legend
- Add a legend below the color axis (normalized PSD)
- Lockdown phase 1b

**Figure 7:**
- Again I have the impression this is a low resolution figure, but it may be related to the pdf conversion. Please check.
- as said above, refer to the method how these graphs were computed.

**Figure 10:**
- please increase font of the legend

**Fig. S2:**
- what happened on 2020-06-24?

**References:**
Nowhere in the paper, the seismic networks are cited. Please do as below:

**CA**: Institut Cartogràfic I Geològic De Catalunya, Institut d'Estudis Catalans (1984). Catalan Seismic Network [Data set]. International Federation of Digital Seismograph Networks. https://doi.org/10.7914/SN/CA

**YS**: Diaz, J., and Schimmel, M. (2019). SANIMS [Data set]. International Federation

of Digital Seismograph Networks. https://doi.org/10.7914/SN/YS$_2$019

**AM**: (1) Raspberry Shake Community; (2) OSOP, S.A.; (3) Gempa GmbH. (2016). Raspberry Shake. (1) OSOP, S.A.; (2) gempa GmbH. https://doi.org/10.7914/SN/AM

---

## Author Comment (AC1) · 3 Feb 2021

Please find enclosed our answers ("==>" marks) intercalated with the reviewer's comments. "==>"

The paper "Seismic monitoring of urban activity in Barcelona during COVID-19 lockdown" by Jordi Diaz and co-authors deals with the seismic signature of the lockdown measures as observed by a very dense local seismic network, installed in Barcelona and composed of a fairly wide variety of instruments: broad-band sensors, short-period sensors, accelerometers and Raspberry shake seismometers. The paper is in a very good shape and scientifically sound, it shows very interesting results regarding how the amount of seismic noise reduction, due to the lockdown measures, is strongly site

dependent even in a so dense network and reflects the local human activity variations. In my opinion, the manuscript deserves to be published in Solid-Earth after minor revision.

==> Thanks for these positive comments

**MAJOR POINTS**

- Section 3.1: it is not clear how you identified the frequency band of interest, that is, the band where the seismic signature of lockdown measures is most evident. It is hard to identify it just by looking at the spectrograms. In addition, each spectrogram shows peculiar features, different from the others.

==> We agree that it is difficult to fix the better bandwidth to detect the effects of human activities is seismic data. However, looking at the spectrograms is, in our opinion, the best option, as it provides a visual way to evaluate the band of interest. Having around 20 stations, we have verified that the frequency band showing large variations is not the same for all the sites. We have produced figures as Fig 4 for different bands and finally choose to use the 2-20 Hz period for all the sites, better than choosing the band with the largest variations for each site. Below 2 Hz, the effects of ocean waves and meteorological factors become important and above 20 Hz the subway activity is relevant for stations located near the tunnels. Restricting the bandwidth (e.g. 4-14 Hz) does not result in significant changes in the interpretation of the data.

- Line 175: how did you calculate such mean value, as well as the power values shown in Table 1? By taking into account the whole time series, or by focusing on week-days/daytime?

==> The relative variations reported in Table 1 have been calculated using the values obtained during working hours. This is now stated in the manuscript.

- The "Discussion and conclusions" section does not discuss all the findings of the manuscript, but it mainly focuses on the comparison between seismic data and mobility
information. I suggest to rename the section and write another section, truly discussing all the findings of the paper.

==> Attending also the comments of Reviewer #2, we have now divided the section into a section focused on the comparison between seismic results and mobility ("4 Seismic results and mobility patterns") and a Conclusions section, where the findings of our work are reviewed..

**MINOR POINTS**

- Line 59: What does "CA" indicate? ==> This is the official code for the ICGC seismic network

- Lines 58-72: the sampling rate information is missing for all the instruments. ==> We have now stated the sampling rates used for the different stations (250 sps for the permanent broad-band, 200 sps for the permanent accelerometers, 100 sps for all but one temporary instrument, 50 sps for this last one)

- Line 78-79: please provide further information about the spectral analysis. For instance, did you divide the 30-min-long windows into smaller windows? If so, how long do the smaller windows last? ==> The processing is done using the same parameters as in Lecocq et al 2020, using 30 min windows with a 50% of overlap.

- Line 80: What do you mean when you talk about "spectra dynamic". Sorry, I do not know this term. ==> It refers in fact to the width of the db bins used during the PPSD calculation. This width is selected to be smaller than the default value (0.25 vs 1.0) in order to have more resolution. We have used here a similar phrasing as in the Supporting material document of Lecocq et al 2020.

- Line 95: "(i.e., (Díaz, 2016a)": please remove the double brackets. ==> Done

- Line 108: is "official time" the local time? ==> Yes, we have now used "official local time" to clarify
- Lines 108-109: "the period of high energy period begins earlier": sounds a bit strange... ==> That's absolutely true. We have changed to "the time interval with the highest energy begins earlier."

- Line 126: "Location of the seismic network": actually, the location is not indicated in the table, but rather site descriptions and power amplitude information. ==> We have changed the Table caption to "Site description and differences in power amplitude observed between the different lockdown phases for all the investigated seismic stations" to clarify this point.

- Line 184: "characteristics. (Figures 7e-f)" -> "characteristics (Figures 7e-h)"? ==> Corrected

- Line 234-240: the power level at this station is very very low compared to what is recorded by the other sensors (it is hard to clearly read it, but it seems to be lower than -300 dB). Is it reasonable? Or is there a problem in the instrument? ==> FBR station has a power level around -120 dB, as shown at Figure 7h or in the spectrogram included al Suppl Fig S1

- Line 258-262: this finding is very interesting. However, it is not highlighted in the abstract, neither in the Discussion and conclusions section. ==> As the main focus of this manuscript is the effect of human activities on seismic noise, we will prefer do not to discuss here in detail the relationship with geology. Our plans include writing an independent contribution including HVSR and autocorrelation measurements before and during the COVID-19 lockdown, where this relationship with geology will be developed.

**FIGURES**

- Figure 1: it would be useful to add in the legend information about the symbols used for the seismic stations. In addition, I suggest to use different symbols (or colors) for short-period and Raspberry Shake sensors (as far as I understand they are both indicated by red dots). The font size of the legend is very small, I suggest to make it bigger.
==> We have increased the size of the geological units legend. As suggested, we have included now a legend presenting the color codes used to show the different types of stations. We have now made discriminated the temporary 1D (Raspberryshake) and 3D instrument using red and orange circles. We have also included some geographic labels (Montjuic, Collserola hills, city center...) that may help the reader

- Figure 2: it would be useful to indicate the different phase names in the figure (similar to what has been done in Figures 4 and 5). ==> We have now indicated the lockdown phases in all the panels

- Figure 3: I suggest to increase the font size. In addition, please correct the caption "Data is expressed as dB as dB relative..." -> "Data is expressed as dB relative..." ==> The Figure has been modified by increasing the font size a 20%. The caption has been corrected

- Figure 4: it is really hard to identify the names of the stations associated with each time series. I suggest to increase the font size of the legend, and to sort the names in the legend into descending noise power order. ==> We have modified the figure following these recommendations

- Figure 6: I suggest to increase the font size of the labels surrounding each plot. ==> Done

- Figure 7: I suggest to increase the font size and to indicate the different phase names in the figure (similar to what has been done in Figures 4 and 5). ==> Done

- Figure 8: I suggest to increase the font size of the labels surrounding each plot. In addition, to make the comparison between seismic noise power and geology easier, I suggest to add another subplot with a schematic map showing the main material distribution, such as Paleozoic, Holocene and Pleistocene materials (a sort of simplified version of Figure 1), as well as the location of the places you cite in the text, as the city center, the industrial factory affecting the station R4B31, Montjuic. ==> We have
increased the font size of the coordinates in the frame. However, we would prefer not to include the geology here, as we think that will make the figure too charged. We have added a comment in the figure caption stating that the reader can refer to Fig 1 to compare with geology. Regarding the location of places cited in the text, we have added them to Fig 1

- Figure 9: "Eastern 2019" -> "Easter 2019". ==> Corrected

- Figure 10: I suggest to increase the font size and to indicate the different phase names in the figure (similar to what has been done in Figures 4 and 5). ==> Done
**Discussion** paper

**Fig. 1.** Location plan of the available seismic stations on the geotechnical map of Barcelona (ICGC). The color code for the different types of instruments used is shown at the top-left corner. Road map from ©

---

## Author Comment (AC2) · 3 Feb 2021

Please find enclosed our answers ("==>" marks) intercalated with the reviewer's comments.

The authors present a very nice local study on the lockdown effects in Barcelona observed on the fixed seismometers and (school) Raspberry shakes. In general this is a clear, well-written study which reads fluently. In this review I suggest only minor changes and propose only cosmetic changes and corrections. There is one statement though that the authors did not fully discuss. The authors explain that they see a direct relationship between the geological cover and the seismic amplification and that this can be detected without the need of performing a microzonation or tomographic inver-

sions. It is somewhat difficult to compare the microzonation results with your findings. It would help to add the zoning of Cid et al. (2001) on the maps in Fig. 8 to see correspondence. Also, why is this correspondence there? Because of interference in the subsoil in this frequency band? This needs some more explanation.

==> Thanks for the kind comments. Regarding the discussion on the relationship between geology and noise amplification, we would prefer do not enter too much in details because i) the focus of this manuscript is the effect of human activities on seismic noise, not really the geology of Barcelona subsoil. Ii) our plans include writing an independent contribution, where HVSR and autocorrelation measurements performed before and during the COVID-19 lockdown will be used to analyze the noise amplification variations. In fact, the relation between amplification and subsoil appears quite direct, with high amplification in the sedimentary zone and low amplification in the Paleozoic outcrops surrounding the city, as commented in the text (lines 254-255). We have now clarified that this is a rather usual feature, documented at larger scales and taken into consideration in seismic risk studies; "This correlation between geology and seismic amplification is a well-known feature that has been documented at the scale of the Iberian Peninsula, where high seismic noise is observed over the sedimentary basin and minimum values over hard-rock regions (Custódio et al., 2014)."

Minor comments:

L22: ambient noise studies: this is a bit vague? What purpose have these ambient noise studies? ==> We have changed the sentence to "...the feasibility of ambient noise studies based on tomographic and interferometric methods in urban environments" to provide more information. However, we think that developing the general objectives of the SANIMS project is not convenient here.

L31: change brackets to Lecocq et al. (2020); same for L34: Lindsey et al. (2020) + Check the rest of the paper. Authors are often within the brackets when they are an active part of the sentence. I refer to this comment by writing "brackets" in this review.

==> Sorry for this "bracket" problem, related to the use of Mendeley. We have now edited the citations along the manuscript.

L34-35: "However, as far as we know, there are no studies available that explore variations in seismic noise within a large city with a space between sites on the order of 2-3 km." Be careful here: you mean that really no-one studied ambient noise variations? There are a lot of geophysical papers about measuring noise in a city, but these are mostly used for array processing and subsurface identification, nevertheless they use the noise variations. ==> No, what I mean is that there are not (to my knowledge) other seismic noise studies related to the COVID19 pandemic using a dense seismic array within a city. In order to make this point clear, we have modified the sentence to "However, as far as we know, there are no studies available that explore variations in seismic noise within a large city during the COVID-19 lockdown period with a space between sites on the order of 2-3 km. "

L43. Thursday the 13th of March L45. Sunday 15 March or Sunday 15th of March In this paper be consistent how you refer to dates. I already noticed 3 different notations. So use one consistent notation: e.g. Sunday 15 March, Sunday 15th of March, Sunday, March 15 (probably the last is the best choice, as you continue using this notation, e.g. March 29, May 4, etc. . .) ==> We have now unified the references to the notation: Sunday, March 15

L67: Is there an official Raspberry shake paper to refer to instead of using the weblink? + It would not harm to cite few city-context papers where Raspberry Shake was already successfully used (e.g. Anthony et al 2018, SRL; De Plaen et al. 2020 (this special issue volume)) ==> Following the reviewer recommendation, we have now used the Anthony et al 2018 paper as a reference for the Raspberryshake instrument, as it provides technical testing of the instrument. We have added a new sentence to highlight the utility of Raspberryshakes in both scientific and outreach objectives, including the De Plaen citation, as well as Subedi et al 2020 (seismometers Nepal schools): "These instruments, designed primarily for "amateur seismologist" users, have proved

to be useful for research projects interested in acquiring quality data and addressing outreach objectives (Plaen et al., 2020; Subedi et al., 2020)."

L69: outside the municipality ==> Corrected

L71: "most recent terranes near the sea": give a geological time frame. Holocene? Pleistocene? L71: Is "materials" a correct term? hard rock? ==> We have modified the sentence to better describe the geologic zones: "The geometry of the network has been chosen to sample the main geological units of the Barcelona area, with sites installed in the Holocene terranes near the sea, above the Pleistocene sediments of the Barcelona plain and in the hills where Paleozoic rocks outcrops (Figure 1)."

L89: "brackets" ==> Done

L95: "brackets" ==> Done

L96: a sentence is missing to link L95 and L96, something like: "because not all stations show reduction up to 45Hz, we will only focus our research..." ==> We have rephrased to "Although some of the stations also show a significant reduction in the 35-45 Hz band, this is not a general feature. Therefore, we have focused our analysis on the seismic signals within 2-20 Hz band, calculating the averaged noise power within this band."

L106: although some of the stations ==> Done

L117: "its hours of completion": weird phrasing. If you refer to the activities, it should be "their hours of completion"; not sure if completion is the correct word to use. ==> Changed to "their ending times"

L119: " a curious observation": is it "curious" when you explain what it is? Perhaps use "remarkable" or "notable", . . . ? ==> We used "curious" to mean that this football-related noise effect can be seen as quite funny. However, we agree that using "remarkable" is better

Table 1: This table should be self-standing. Explain what the pre/lock1, pre/lock2 , pre/new norm columns stand for. ==> We have now completed the Table caption

L135: "clearly identified": try to minimize the use of words as "clearly, obvious" etc, rather explain why it's clear. E.g. a drop follows an lockdown phase, a mean rise follows a phase, etc. . . ==> We have rephrased to "For most of the stations, each lockdown phase has a well-differentiated mean amplitude value"

L139: '90dB = -90 dB ? ==> Corrected

L150: clearly is ok in this phrase ==> Ok

L155: nearest neighbor algorithm ==> Corrected

L170: here again reference should be made to the technique how these figures were computed. Or this can be mentioned in the caption of Fig. 7 ==> We have modified the sentence to "Figure 7 shows the power amplitude in the 2-20 Hz band measured every 15 minutes (light blue line) and the daily mean value during business hours (dark blue line) for some selected sites"

L175: indicate which stations you are talking about by e.g. putting station names in brackets behind "in schools" ==> We have added the list of the stations in school, although the information was already available at Table 1

L186: (2017) "brackets" ==> Done

L239: "brackets" ==> Done

L252 : for people that never visited Barcelona, indicate where montjuic is located ==> A label indicating the location has now been added to Figure 1

L253: how are both maps consistent? Please explain. Do the 4 zones correspond to the microzonation map of Cid2001? Perhaps it would be interesting to put the zonation boundaries on the map. ==> (see the answers to general comment and comment "L358" below)

L254: "brackets" ==> Done

L255: "brackets" ==> Done

L280: "probably be related to civil works near to the station site." This is speculation. Needed in this paper? You can just say this increase remains unexplained due to a lack of local site/communal information. ==> We have modified the sentence following the reviewer's recommendation

L358: I think you can separate the discussion (everything before L358) and the conclusion (starting from L358) ==> Attending also the recommendations of Reviewer # 1, we have now split the former section in two parts: "4 Seismic data and mobility patterns " and "5 Conclusions".

L358: in the discussion, a discussion on the comparison between your findings and microzonation findings is lacking. Is this anywhere else observed? Can this be done in other cities (check e.g. papers in this special issue). If this is new, it should be mentioned in the conclusions, and perhaps in the abstract. ==> (see also the answer to the general comments above). We have reworked the paragraph to clarify that the lockdown period allows suppressing the perturbations due to local sources, making the relationship between geology and seismic noise clearer than in regular times.

L360: "brackets" ==> done

L373: I think it's "near real-time", not nearly real-time. Please check. ==> I think both terms can be used, but "near real-time" appear to be more common. Changed

L401-403: add these cited references to the reference list and follow the proper citation rules in these lines: i.e. Wessel et al. (2013), Lecocq et al. 2020. T. Lecocq, F. Massin, C. Satriano, M. Vanstone, T. Megies, SeismoRMS - A simple Python/Jupyter Notebook package for studying seismic noise changes, Version 1.0, Zenodo (2020); doi:10.5281/zenodo.3820046 ==> The references have now been included

L437: wrong author list: either provide full names of all 75 authors or write: Lecocq,

T., Hicks, S. P., Van Noten, K., Van Wijk, K., Koelemeijer, P., et al. : Global quieting of high-frequency seismic noise due to COVID-19 pandemic lockdown measures, Science (80)., 369(11 September), 1338–1343, 2020. ==> Corrected

Figure 1: - The colors of the geology in the background of the Holocene, Pleistocene, Pliocene and Carboniferous periods are very difficult to distinguish. ==> The colors for the geological units come directly from the WCS server of the ICGC and are those commonly used to refer to those units. In order to make them clearer, we have now modified the color used for roads and streets. - Add coordinates to the figure. ==> We have added coordinates, without using frames to keep the figure clear - A little inset with the locality of Barcelona indicate on the scale of Spain would be of interest for the international reader. ==> The proposed inset has now been added - can you use a different symbol for Seismometers and Accelerometers (or Raspberryshake)? ==> Attending also to the observation of Reviewer #1, the stations are now represented using different colors - caption: "The dark and light blue ==> Corrected

Figure 2: in the caption, please write again the lockdown phase dates that correspond to the 3 dashed lines. ==> Done

Figure 3: - indicate when lockdown happened with a vertical bar on this figure - add to caption: Weekdays are indicated in green. - add to caption: Trends in noise variation prior to lockdown. - explain again where the stations are located (e.g. AM.R888C in school, . . .) ==> We have modified the legend to: "Trends in power acceleration variation in the 2-20 Hz band prior to lockdown at the ICJA (broad-band), R888C, R4B31 (short period stations installed in high schools) and BAJU (accelerometer located downtown). Data is expressed as dB relative to 1 m2/s4/Hz. Red line marks the beginning of the lockdown period. Green shading indicates weekdays. - What are the spikes on CA.BAJU ? ==> As stated in the text (l 269) "The spikes often observed for the BAIN and BAJU sites are related to data transmission problems and are not relevant for the discussion" - Refer to Lecocq's seismo RMS technique either in the text (L102) or in the caption, to highlight how you made this figure: T. Lecocq, F.

Massin, C. Satriano, M. Vanstone, T. Megies, SeismoRMS - A simple Python/Jupyter Notebook package for studying seismic noise changes, Version 1.0, Zenodo (2020); .doi:10.5281/zenodo.3820046 ==> We have added this reference in "Data and processing" section (lines 79-81): "Data processing is based on the "SeismoRMS" software package, publicly available on Github (Lecocq et al. 2002a)." A reference to the software package was already included in the Acknowledgments section.

Figure 4: It would be more intuitive if the legend could be ordered according to the observations: so MTJR on top (dark green), then R4B31 (light green), etc. . . this would easier to read the diagram. ==> The figure has now been modified following this recommendation

Figure 6: - This figure is of low resolution. Can you increase the resolution. - Also the coordinates are unreadable - Topographical contours are included, but height is not shown in the legend - Add a legend below the color axis (normalized PSD) - Lockdown phase 1b ==> Regarding resolution, the original svg file appears fine. We think that the problem may be related to the insertion of the figure in the Word document. We have now increased the size of the coordinate labels and added the color bar legend. Regarding topographical contours, we precise now in the legend that thick lines are for 100m isolines and thin ones for 50m. We prefer do not to include the map for Lockdown phase 1b, as is very similar to that of Phase 1 and will surcharge the Figure. The reader can find it at Supp. Figure S2

Figure 7: - Again I have the impression this is a low resolution figure, but it may be related to the pdf conversion. Please check. - as said above, refer to the method how these graphs were computed. Regarding resolution, the svg file appears fine ==> As commented before, an explicit reference to SeismoRMS is now included in the "Data and Processing section". Most of the figures (figs, 3,4,5,7,9,10) are derived from this software; we think that including a reference at each figure will be reiterative and will not provide further information to the reader.

Figure 10: - please increase font of the legend ==> Done

Fig. S2: - what happened on 2020-06-24? ==> 24th June is the St-Jean celebration, a bank holiday. Usually, the night before there are large celebrations around the city, including fireworks, music etc. . .

References: Nowhere in the paper, the seismic networks are cited. Please do as below: CA: Institut Cartogràfic I Geològic De Catalunya, Institut d'Estudis Catalans (1984). Catalan Seismic Network [Data set]. International Federation of Digital Seismograph Networks. https://doi.org/10.7914/SN/CA YS: Diaz, J., and Schimmel, M. (2019). SANIMS [Data set]. International Federation of Digital Seismograph Networks. https://doi.org/10.7914/SN/YS2019 AM: (1) Raspberry Shake Community; (2) OSOP, S.A.; (3) Gempa GmbH. (2016). ==> The network references have now been added

[Figure]

**Fig. 1.** Location plan of the available seismic stations on the geotechnical map of Barcelona (ICGC). The color code for the different types of instruments used is shown at the top-left corner. Road map from ©

[Figure]

**Fig. 2.** Spectrograms for period from February 24 to July 14, 2020 corresponding to the ICJA, RBE49, R4B31 and BAJU sites. The solid lines mark the beginning of the lockdown period and the dashed lines show th

[Figure]

**Fig. 3.** Trends in power acceleration variation in the 2-20 Hz band prior to lockdown at the ICJA (broad-band), R888C, R4B31 (short period stations installed in high schools) and BAJU (accelerometer located do

[Figure]

**Fig. 4.** Variation of the power of the seismic acceleration in the 2-20 Hz band for all the investigated sites, expressed in dB. The colored lines show the daily average during business hours. The vertical bar

[Figure]

**Fig. 5.** Normalized power in the 2-20 Hz band during working hours for all the stations (gray lines) and the corresponding mean profile (blue line). Bars as in Figure 4.

[Figure]

**Fig. 6.** Daily maps representing the normalized power in the 2-20 Hz band during working hours. Each image corresponds to a Monday. (Supplementary figure S2 shows all the daily snapshots). a) Pre-lockdown peri

[Figure]

[Figure]

**Fig. 7.** Variations of the PSD of the seismic acceleration in the 2-20 Hz band for representative stations. a-d) sites that follow the general trend. e-h) sites with particular characteristics (see text). Ligh

[Figure]

**Fig. 8.** Daily maps representing the real power values in the 2-20 Hz band during working hours. Each image corresponds to a Monday. (Supplementary figure S4 shows all the daily snapshots). a) Pre-lockdown per

[Figure]

**Fig. 9.** Long term (1/1/2019 – 30/9/2020) PSD of the seismic accelerations in the 2-20 Hz band. a) ICJA broad-band seismometer. b) BAJU accelerometer. c) BAIN accelerometer. Color code as in Figure 7.

[Figure]

[Figure]

[Figure]

[Figure]

**Fig. 10.** Mobility data from different sources compared to seismic data. a) Normalized mean seismic power during business hours (9:00 – 19:00) vs. mobility data from Apple and Google. b) Normalized mean seismic